# Taxonomic Description and Complete Genome Sequencing of *Pseudomonas silvicola* sp. nov. Isolated from *Cunninghamia laceolata*

Longyan Tian [1] , Yanfeng Zhang [2], Hua Yang [1], Qian Zhao [1], Hualong Qiu [1], Jinzhu Xu [1,*] and Changsheng Qin [1,*]

1   Guangdong Provincial Key Laboratory of Silviculture, Protection and Utilization, Guangdong Academy of Forestry, Guangzhou 510520, China; tianlongyan@sinogaf.cn (L.T.); yanghua@sinogaf.cn (H.Y.); zhaoqian202305@163.com (Q.Z.); hlqiu@sinogaf.cn (H.Q.)
2   Guangdong Eco-Engineering Polytechnic, Guangzhou 510520, China; zhangyanfeng_plant@163.com
*   Correspondence: xujinzhu@sinogaf.cn (J.X.); qincs@sinogaf.cn (C.Q.)

**Abstract:** The *Pseudomonas* strain T1-3-2$^T$ isolated from the cone of *Cunninghamia laceolata* exhibited growth-promoting and antifungal activity. Strain T1-3-2$^T$ was characterized by a polyphasic taxonomy and complete genome sequencing analysis to explore its taxonomic position and biocontrol potentials fully. The results revealed that strain T1-3-2$^T$ shares a high degree of similarity with *Pseudomonas eucalypticola* and is distinct from any known *Pseudomonas* species. The G + C content was 61.65%, and the difference was greater than 1 compared to "*P. eucalypticola*". Additionally, values of the average nucleotide identity blast (ANIb), average nucleotide identity MUMmer (ANIm), and DNA-DNA hybridization (DDH) between T1-3-2$^T$ and its closest known related species, "*P. eucalypticola*", were below the thresholds necessary for species delineation. Furthermore, the T1-3-2$^T$ strain exhibited the distinctions with the multiple polar flagella and the specific quinone system with MK8 compared with that of "*P. eucalypticola*". Collectively, these findings affirm the designation of strain T1-3-2$^T$ as a new *Pseudomonas* species, proposed to be named *Pseudomonas silvicola*, with T1-3-2$^T$ as the type strain. Genomic analyses revealed strain T1-3-2$^T$ contains three circular DNA contigs, including a 7,613,303 bp chromosome and two plasmids (952,764 bp and 84,880 bp). Bioinformatics analyses further offered potential insight into the molecular mechanisms whereby this strain can promote plant growth and control disease, revealing encoded genes related to antibiotic and secondary metabolite production, the uptake and biosynthesis of siderophores, and pyoverdine biosynthesis. These genomic data offer a valuable foundation for future efforts to apply the T1-3-2$^T$ strain in research contexts.

**Keywords:** *Pseudomonas*; taxonomic description; genome



## 1. Introduction

The *Pseudomonas* family, comprised of motile, nonsporulating, rod-shaped γ-proteobacteria, is noteworthy for encompassing the most extensive assortment of acknowledged gram-negative bacterial species [1,2]. They shared some features in common, such as aerobic growth, catalase- and oxidase-positive, with the chemotaxonomic characteristics containing a fatty acid profile with $C_{16:0}$, summed feature 3 ($C_{16:1}\omega7c$ and/or $C_{16:1}\omega6c$), summed feature 8 ($C_{18:1}\omega7c$ and/or $C_{18:1}\omega6c$), the hydroxylated fatty acid $C_{10:0}3$-OH, quinone system ubiquinone Q-9, and a polyamine pattern with the predominant amines putrescine and spermidine [3]. The advent of phylogenomics has enabled the more accurate taxonomic classification of members of the *Pseudomonas* genus, leading to a significant increase in the number of documented species [2]. Over 313 *Pseudomonas* species have been published with correct name under the ICNP, with 106 of these species published since 1 January 2021 (http://www.bacterio.net/pseudomonas.html, accessed on 30 March 2023) [4].

The *Pseudomonas* genus is highly diverse and includes members that inhibit a range of holobionts with substantial metabolic versatility [5–7]. Many beneficial plant-associated *Pseudomonas* species have been documented to date, with this being one of the best-characterized groups of bacteria with desirable plant growth promotion (PGP) and antagonist properties [8]. Complete genome sequencing has revealed PGP *Pseudomonas* (PGPP) strains, including *P. fluorescens* strains PICF [9] and BRZ63 [10], the *P. chlororaphis* strains GP72 [11] and PCL1606 [12], and *P. oryziphila* 1257 [7], to offer substantial genetic promise as potential biocontrol agents. The ability of PGPP species to directly promote plant growth can be achieved by altering phytohormone levels within these plants, stimulating ion transport, and/or increasing nutrient availability [13,14]. Moreover, the indirect promotion of plant growth by these species can be accomplished by inhibiting phytopathogen growth via antimicrobial compound production, competitive mechanisms, and/or activating defense responses within plants, such as the engagement of more robust systemic resistance [15,16].

*Cunninghamia lanceolate* (Lamb.) Hook, a popular species of rapidly growing tree with large-scale planting areas, is prone to disease outbreaks with fungal pathogens, which often leads to substantial tree damage and loss in China [17]. Over the past three decades, a series of biocontrol agents for the disease *Cunninghamia lanceolate* have been reported, including fungal strains *Gliocladium* virens [18] and *Trichoderma* harzianumin [19], *Bacillus* species [20], *B.* lichenformis [21], and *B. atrophaeus* [22]. In our previous study, we found *Pseudomonas* strain T1-3-2 exhibited plant growth promotion (PGP) and antifungal properties against phytopathogens of *C. lanceolate* [23], forming a monophyletic group with *Pseudomonas eucalypticola* based on 16s rRNA phylogenetic analyses. However, its taxonomic classification and potential utility were not clarified, and this hindered its fundamental research and utilization. This present study was designed to conduct polyphasic taxonomic analysis and genome-based studies of this strain. Overall, these analyses revealed strain T1-3-2$^T$ to be a novel *Pseudomonas* species, with T1-3-2$^T$ (CP093280$^T$) as the proposed type strain, with promising potential as a biocontrol agent. The findings not only expand our understanding of the strain T1-3-2$^T$ but also enhance the diversity of the *Pseudomonas* family. Furthermore, this strain could be harnessed for disease biocontrol in *C. lanceolate*.

## 2. Results and Discussion

### 2.1. The 16s rRNA and Multilocus Sequence Analysis (MLSA-)-Based Phylogenetic Analyses

A 1396 bp 16s rRNA gene fragment was amplified from strain T1-3-2$^T$, and the sequence was deposited in GenBank (accession number: OM920535). Then a similarity search was performed with this sequence in EzBioCloud, revealing that 54 strains exhibited >97.71% pairwise similarity with T1-3-2$^T$. Of these, 5 species shared >99.00% similarity, including *Pseudomonas vancouverensis*, "*Pseudomonas eucalypticola*", *Pseudomonas moorei*, *Pseudomonas izuensis*, and *Pseudomonas koreensis,* in 34 validly named species. Subsequently, a similar phylogenetic tree was constructed based on the 16S rRNA gene sequences shown in Figure 1. Strain T1-3-2$^T$ formed a clade with a bootstrap value of 99%, and both of these strains were located within a separate cluster along with "*P. eucalypticola*". Additionally, a similar phylogenetic tree was exhibited based on a 4000 bp concatenated sequence of the partial 16S rRNA, gyrB, rpoB, and rpoD genes (Supplementary Figure S1). This analysis revealed strains T1-3-2$^T$ to be members of the same species, exhibiting a high degree of similarity to "*P. eucalypticola*" NP-1$^T$ with a bootstrap value of 100%. The sequences of the *Pseudomonas* type strains used in 16s rRNA phylogenetic analysis and MLSA phylogenetic analysis are shown in Supplementary Tables S1 and S2, respectively.

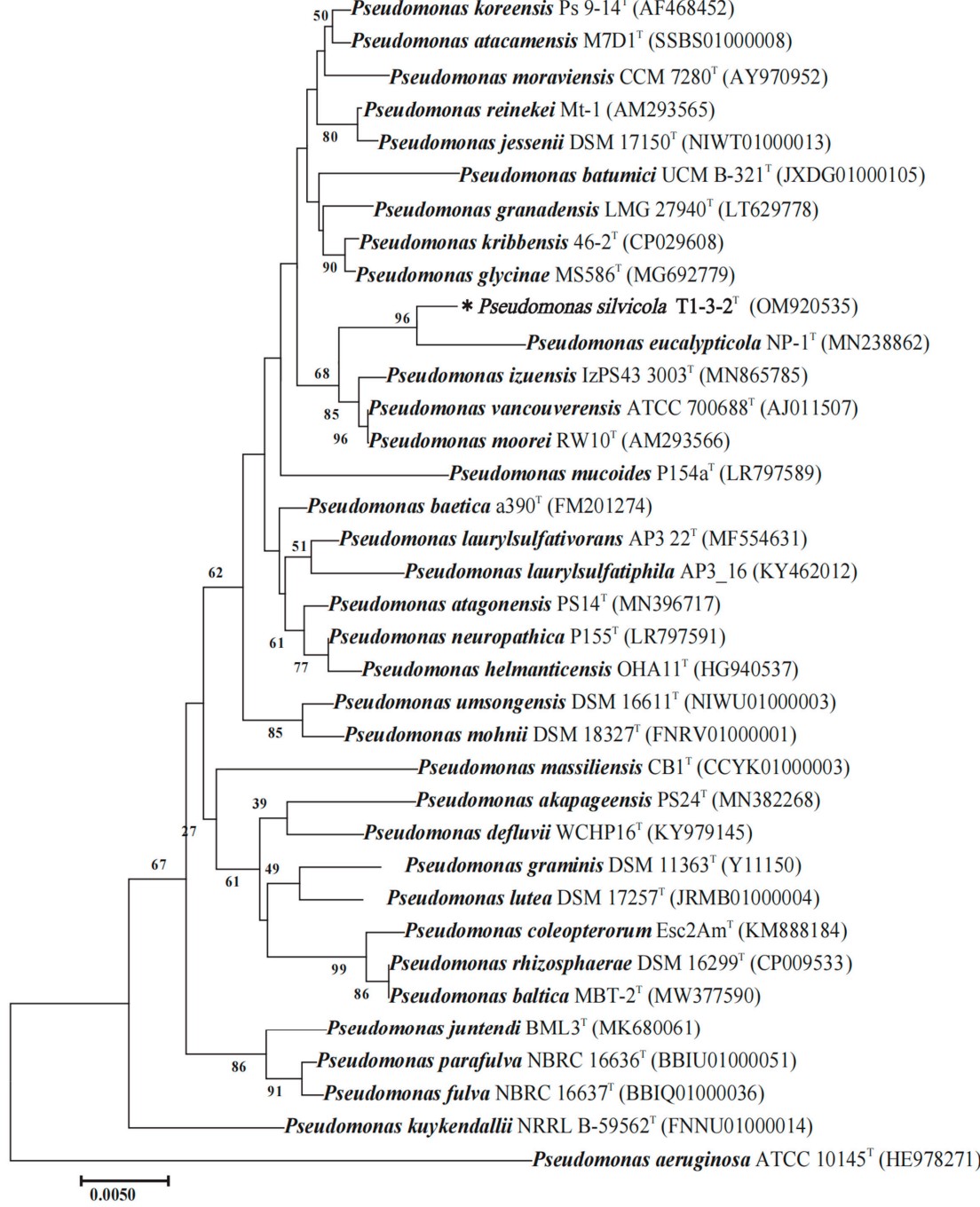

**Figure 1.** Phylogenetic tree based on 16S rRNA of strain T1-3-2$^T$ and related type strains of *Pseudomonas* species. The evolutionary history was inferred using the neighbor-joining method. The optimal tree with the sum of branch length = 0.19417617 is shown. The percentage of replicate trees in which the associated taxa clustered together in the bootstrap test (2000 replicates) is shown next to the branches. The tree is drawn to scale, with branch lengths in the same units as those of the evolutionary distances used to infer the phylogenetic tree. The evolutionary distances were computed using the Jukes–Cantor method and are in units of the number of base substitutions per site. The analysis involved 37 nucleotide sequences. All ambiguous positions were removed for each sequence pair. There were a total of 1463 positions in the final dataset. Evolutionary analyses were conducted in MEGA7. The name and isolate number of T1-3-2$^T$ were indicated with an *.

## 2.2. Analyses of DNA G + C Content, DNA Relatedness, and Phylogenomic Analyses

The G + C content of the T1-3-2[T] was 61.65 mol%. It was within the range (58–69 mol%) of *Pseudomonas*, but with a difference greater than one than that of "*P. eucalypticola*" (63.96 mol%), which showed a G + C mol% difference in different species [24]. Though the tetranucleotide signature (TETRA) frequency of T1-3-2[T] compared to "*P. eucalypticola*" NP-1[T], was 0.99277, this value was higher than the 0.99 TETRA signature value recommended for species delineation. Taxonomic classification via the ANI and in silico DDH approaches revealed that T1-3-2[T] exhibited ANIb (90.3%), ANIm (92.80%), and DDH values (48.2%) relative to "*P. eucalypticola*" NP-1[T] (Table 1) that were below the generally accepted thresholds for the delineation of species, as members of the same species generally share 95–96% ANI and 70% DDH [25,26].

**Table 1.** Average nucleotide identity (ANI) and in silico DNA–DNA hybridization (DDH) comparisons between T1-3-2[T] and its closely related *Pseudomonas* spp.

| Genome [a] / Summary for T1-3-2[T] | rANI Value and Aligned Percentage [%] [b] | | Tetra [%] [c] | DDH [%] [d] | G + C mol% Difference [e] |
|---|---|---|---|---|---|
| | ANIb | ANIm | | | |
| *Pseudomonas eucalypticola* NP-1[T] GCF_013374995 | 90.3 [66.5] | 92.8 [64.5] | 99.3 | 48.2 | 1.48 |
| *Pseudomonas rhizosphaerae* DSM 16299[T] GCA_000761155 | 76.9 [35.7] | 85.4 [17.8] | 95.6 | 23.8 | 0.35 |
| *Pseudomonas coleopterorum* LMG 28558[T] GCA_900105555 | 76.5 [37.1] | 85.4 [17.6] | 95.5 | 23.6 | 0.38 |

(a) Accession numbers referring to genome sequences used for analysis are provided in Table S3. (b) The same species share 95–96% of ANI. ANIb is short for ANI-Blast. ANIm is short for ANI-MUMmer. (c) The same species share 99% of Tetra. (d) The same species share at least 70% in silico DDH. The accession numbers of "*P. eucalypticola*", *P. coleopterorum*, and *P. rhizosphaerae* are used in DDH in GenBank, respectively. DDH [%] results were based on recommended Formula (2). (e) The same species share at least 70% in silico DDH.

Further, phylogenomic trees were constructed with the type (strain) genome server (TYGS) using the reference type strains and genome sources compiled in Supplementary Table S3. This analysis revealed this strain to be located on an independent branch supported by a 100% bootstrap value and clustered with "*P. eucalypticola*" NP-1[T] (Figure 2), with the differences in these branch lengths further supporting the differences in evolutionary rates for these two strains.

## 2.3. Morphological, Physiological, and Biochemical Characterization

Colonies of T1-3-2[T] were beige and round with smooth edges on an LB medium (Figure 3A), while they had an inky purple color and were irregular with smooth edges on a TYB medium after 48 h of inoculation at 25 °C (Figure 3B). Colonies were beige to reddish-brown and round with smooth edges after growth for three days on a PDA medium (Figure 3C). These morphology and colonial appearances were similar to "*P. eucalypticola*" NP-1[T]. Cells of this species are gram-negative nonsporulating rods ~0.6–0.9 × 1.8–2.6 μm in size, with the motility of these microbes being facilitated by multiple polar flagella ~2.0–5.2 μm in length (Figure 3D,E and Table 2). In contrast to the "*P. eucalypticola*" NP-1[T] that was rod-shaped (1.0 μm wide, 2.0 μm length averages) and motile by a single polar flagellum [27].

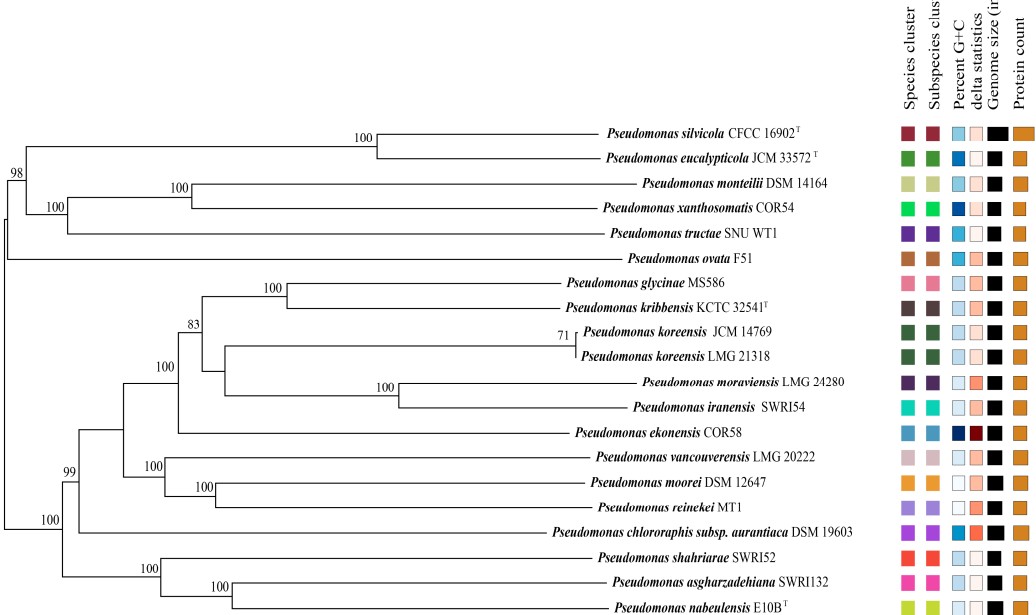

**Figure 2.** Phylogenomic tree of strain T1-3-2[T] and related type strains of the genus *Pseudomonas* are available on the TYGS database. Tree inferred with FastME 2.1.6.1 from the GBDP distances calculated from genome sequences. The branch lengths are scaled in terms of GBDP distance Formula d$_5$. The numbers above branches are GBDP pseudobootstrap support values >60% from 100 replications, with an average branch support of 91.2%. The tree was rooted at its midpoint.

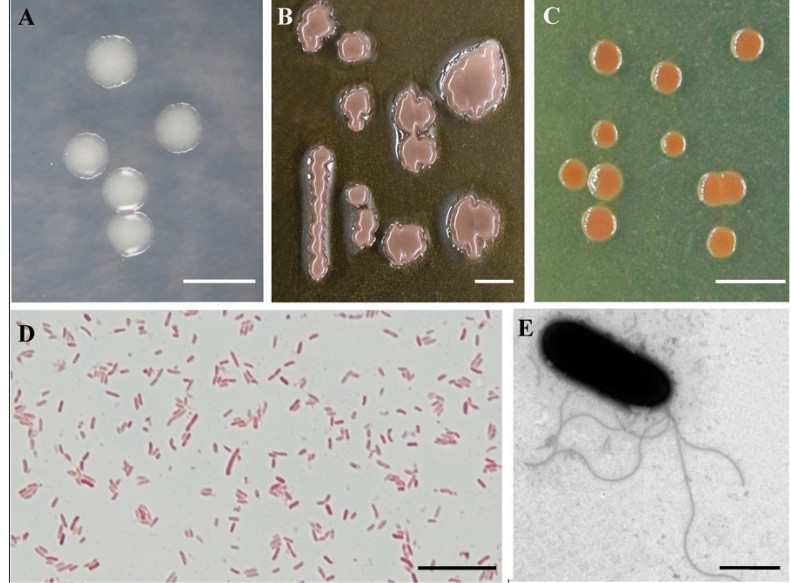

**Figure 3.** The morphology of *P. silvicola* T1-3-2[T]. Colonies of *P. silvicola* T1-3-2[T] grown on LB, TYB, and PDA medium at 25 °C for 48 h. Bar= 5 mm (**A–C**). The bacterial morphology was obtained by microscopy after gram staining. Bar = 10 μm (**D**). The multiple polar flagella were obtained by transmission electron microscopy. Bar = 1 μm (**E**).

**Table 2.** Phenotypic characteristics of *P. silvicola* T1-3-2$^T$ and "*P. eucalypticola*" NP-1$^T$.

| Characteristic | T1-3-2$^T$ | NP-1$^T$ | Characteristic | T1-3-2$^T$ | NP-1$^T$ |
|---|---|---|---|---|---|
| Bacterial morphology | rod-shaped | rod-shaped | Gram staining | - | - |
| Size (µm) | 0.6–0.9 × 1.8–2.6 | 1.0 × 2.0 | Fluorescent pigments | + | - |
| Number of polar flagella | 3–5 | 1 | O-F test | O/- | O/- |
| Oxidase | - | - | Hydrolysis of gelatin | + | - |
| Catalase | + | + | Hydrolysis of starch | + | + |
| Nitrate reduction | + | - | Hydrolysis of DNA | - | - |
| Arginine dihydrolase | + | + | Citrate utilization | + | + |
| Acetamidase hydrolysis | - | | D-fructose | w | + |
| dextrin | - | + | D-galactose | + | + |
| D-mannose | + | + | gentiobiose | w | + |
| D-Mannose | w | + | α-D-glucose | + | + |
| L-Rhamnose | + | + | D-mannitol | w | + |
| Quinic Acid | + | + | L-Aspartic Acid | + | + |
| Bromo-Succinic Acid | w | + | D-trehalose | w | + |
| Glucuronamide | + | + | formic acid | w | + |
| D-galacturonic acid | + | + | D-Gluconic Acid | + | + |
| α-keto glutaric acid | - | w | D-saccharic acid | + | w |
| methyl pyruvate | - | w | D-turanose | - | w |
| Growth at temperature °C | 10–37 | 4–37 | Maltose | - | - |
| NaCl (%, *w/v*) | 0–2 | 0–2 | pH | 5–8 | 3–7 |

*2.4. Chemotaxonomic Characterization*

　　As an important chemical characteristic for bacterial identification, cellular fatty acid analysis of T1-3-2$^T$ and NP-1$^T$ was performed next (Table 3). The most abundant fatty acids present in strain T1-3-2$^T$ included hexadecanoic acid ($C_{16:0}$: 25.08%), 17-carbon cyclopropane fatty acid ($C_{17:0}$ cyclo: 17.64%), summed feature 8 ($C_{18:1}ω7c/C_{18:1}ω6c$: 14.71%), and summed feature 3 ($C_{16:1}ω7c/C_{16:1}ω6c$: 10.13%). These major compound results, as well as the levels of $C_{10:0}$ 3-OH (5.96%), $C_{12:0}$ 3OH (5.64%), $C_{19:0}$ cyclo w8c (5.51%), and $C_{12:0}$ 2OH (5.39%), are in line with the results observed for NP-1$^T$, forming a fatty acid profile characteristic of group I of *Pseudomonas* strains that exhibit $C_{10:0}$ 3-OH and $C_{12:0}$ 3-OH [28]. Strains T1-3-2$^T$ and NP-1$^T$ also contained detectable $C_{16:0}$ 3-OH, and these cellular fatty acid profiles for strains T1-3-2$^T$ and NP-1$^T$ were distinct from those of any known groups. Strain T1-3-2$^T$ also lacked certain fatty acids, including $C_{20:2}$ w6,9c, $C_{19:0}$ iso, $C_{14:0}$ 2OH, $C_{15:0}$ 3OH, and summed feature 7 when compared to the NP-1$^T$ strain (Table 3). The predominant respiratory quinone of strain T1-3-2$^T$ was Q-9, typical of the genus *Pseudomonas* [3,28]. Differently, T1-3-2$^T$ was of methylnaphthoquinone MK8, a specific quinone system compared to that of "*P. eucalypticola*". These data suggest that strain T1-3-2$^T$ had a fatty acid profile distinct from that of NP-1$^T$, further supporting its identity as a novel species in the *Pseudomonas* genus.

**Table 3.** Chemotaxonomic characterization of *P. silvicola* sp. nov. and the closely related species in *Pseudomonas* genus.

| Chemotaxonomic Characterization | | Peak Name | T1-3-2T Percent Named% | NP-1T Percent Named% |
|---|---|---|---|---|
| Fatty acid | Straight-chain fatty acids | 10:0 | 0.99 | 2.57 |
| | | 12:0 | 3.61 | 4.39 |
| | | 14:0 | 0.31 | 0.19 |
| | | 16:0 | 25.08 | 19.67 |
| | | 17:0 | 0.11 | 0.17 |
| | | 18:0 | 0.86 | 0.7 |
| | | 19:0 | 0.15 | 0.13 |
| | Unsaturated fatty acids | 17:1 w7c | 0.11 | 0.15 |
| | | 18:1 w7c 11-methyl | 0.16 | 0.19 |
| | | 20:2 w6,9c | - | 0.07 |
| | Branched fatty acid | 19:0 iso | - | 0.06 |
| | Hydroxy fatty acid | 10:0 3OH | 5.96 | 11.15 |
| | | 12:0 2OH | 5.39 | 5.13 |
| | | 12:1 3OH | 2.76 | 6.6 |
| | | 12:0 3OH | 5.64 | 7.16 |
| | | 14:0 2OH | - | 0.17 |
| | | 15:0 3OH | - | 0.05 |
| | | 16:0 3OH | 0.12 | 0.25 |
| | Cyclopropane acids | 17:0 cyclo | 17.64 | 13.56 |
| | | 19:0 cyclo w8c | 5.51 | 6.97 |
| | Summed feature | Summed feature 2 | 0.44 | 0.72 |
| | | Summed feature 3 | 10.13 | 7.58 |
| | | Summed feature 5 | 0.27 | 0.21 |
| | | Summed feature 7 | - | 0.11 |
| | | Summed feature 8 | 14.71 | 12.06 |
| | | Summed feature 9 | 0.07 | - |
| Quinone system | Ubiquinone | Q-8 | 15.99 | 34.08 |
| | | Q-9 | 84.02 | 65.92 |
| | Methylnaphthoquinone | MK8 | 100.00 | - |

Note: Only results with amounts higher than 0.5% for at least one strain are presented. Summed feature 3 contained $C_{16:1}$ w7c/$C_{16:1}$ w6c; summed feature 8 contained $C_{18:1}$ w7c/$C_{18:1}$ w6c.

*2.5. Genomic Sequencing and the General Genomic Features of P. silvicola sp. nov.*

In order to investigate the biocontrol potential inherent in the genome properties, the genome of strain T1-3-2[T] was sequenced utilizing Oxford nanopore technologies. This process resulted in the assembly of three high-quality contigs, achieving an average coverage of 203-fold. The circular diagram of the T1-3-2[T] chromosome is visually depicted in Figure 4. In total, 8116 protein-coding sequences (CDSs) were predicted to be encoded on this chromosome, along with 20 genomic islands identified with IslandPath-DIMOB v0.2 [29], 38 pseudogenes, 95 tRNAs, 77 ncRNAs, and 25 rRNA genes (Table 4). PHASTER (http://phaster.ca/, accessed on 12 January 2021) was utilized to identify prophage regions, including three at contig1 (P1, P2, and P3) and one (P4) at contig2. One of these regions was found to be intact (P2, *Pectobacterium* phage ZF40, NCBI accession: NC_019522), while the remaining three were incomplete regions (P1, Stx2-converting phage Stx2a_F451, NC_049924; P3, *Pseudomonas* phage F10, NC_007805; P4, Salmonella phage SJ46, NC_031129) (Table 4). When the identified CDSs on the *P. silvicola* T1-3-2[T] chromosome were annotated using the NR, Pfam, Swiss-Prot, and TrEMBL databases, the respective numbers of retrieved entries were 7887, 6644, 4667, and 4667, respectively (Table 4).

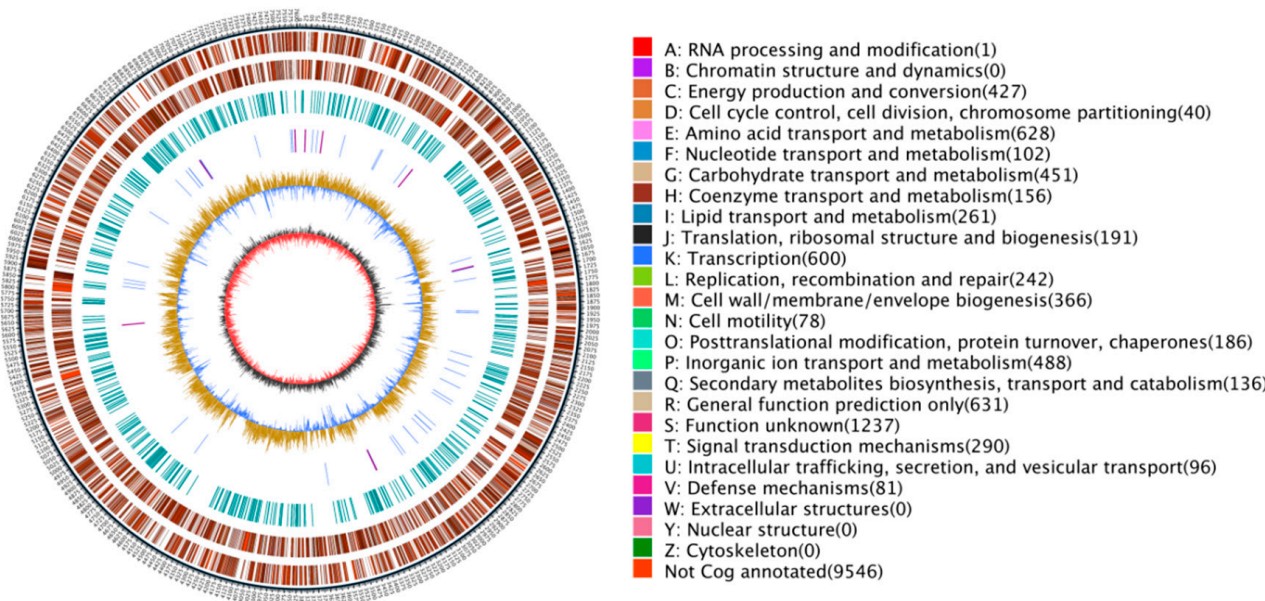

A: RNA processing and modification(1)
B: Chromatin structure and dynamics(0)
C: Energy production and conversion(427)
D: Cell cycle control, cell division, chromosome partitioning(40)
E: Amino acid transport and metabolism(628)
F: Nucleotide transport and metabolism(102)
G: Carbohydrate transport and metabolism(451)
H: Coenzyme transport and metabolism(156)
I: Lipid transport and metabolism(261)
J: Translation, ribosomal structure and biogenesis(191)
K: Transcription(600)
L: Replication, recombination and repair(242)
M: Cell wall/membrane/envelope biogenesis(366)
N: Cell motility(78)
O: Posttranslational modification, protein turnover, chaperones(186)
P: Inorganic ion transport and metabolism(488)
Q: Secondary metabolites biosynthesis, transport and catabolism(136)
R: General function prediction only(631)
S: Function unknown(1237)
T: Signal transduction mechanisms(290)
U: Intracellular trafficking, secretion, and vesicular transport(96)
V: Defense mechanisms(81)
W: Extracellular structures(0)
Y: Nuclear structure(0)
Z: Cytoskeleton(0)
Not Cog annotated(9546)

**Figure 4.** Circularized genome map of *P. silvicola* T1-3-2$^T$. Circle 1, genomic position in kb (total 8,650,947 bp); circles 2 and 3, predicted protein-coding sequences (CDS) on the forward [transcribed clockwise] (outer part) and the reverse [transcribed counterclockwise] (inner part) strand colored according to the assigned COG classes; circle 4, a repeat sequence; circle 5, tRNA and rRNA, blue is tRNA, purple is rRNA; circle 6, G + C content showing deviations from the average. The light-yellow part indicates that the G + C content in this region is higher than the average G + C content in the genome, and the higher the peak value is, the greater the difference is with the average G + C content. The blue part indicates that the G + C content in this region is lower than the average G + C content in the genome; circle 7, G + C skew. Dark gray represents the region with G content greater than C, and red represents the region with C content greater than G.

**Table 4.** Genome features of *P. silvicola* T1-3-2$^T$.

| Attributes | T1-3-2$^T$ |
|---|---|
| Genome size (bp) | 8,650,947 bp |
| G + C content (%) | 61.7 |
| Total genes | 8116 |
| Protein coding genes | 8116 |
| RNAs | 197 |
| tRNAs | 95 |
| Ribosomal RNAs (5S, 16S, 23S) | 9, 8, 8 |
| ncRNAs | 77 |
| Pseudogenes | 38 |
| Genomic island | 20 |
| Prophage | 4 |
| Gene cluster | 15 |
| Genes with predicted functions | |
| Annotated genes databases | 7884 |
| Nonredundant protein database | 7870 |
| eggNOG_Annotation | 6862 |
| Gene Ontology_Annotation | 5779 |
| Kyoto encyclopedia of genes and genomes | 4055 |
| Pfam_Annotation | 6644 |
| Swissprot_Annotation | 4667 |
| TrEMBL_Annotation | 4667 |

## 2.6. Genomic Annotation Using the GO, KEGG, and eggNOG Databases

Next, the eggNOG, GO, and KEGG databases were used to annotate the *P. silvicola* T1-3-2$^T$ CDSs, yielding 6862, 5779, and 4055 respective entries (Table 4). The 6862 annotated genes from *P. silvicola* T1-3-2$^T$ were clustered into 22 COGs (Table S4), with the five most enriched COGs, including amino acid transport and metabolism, general function prediction only, transcription, inorganic ion transport and metabolism, and carbohydrate transport and metabolism, respectively, corresponding to 9.37%, 8.96%, 8.58%, 7.20%, and 6.73% of the annotated genes (Supplementary Figure S2A; Table S4). When these genes were further categorized through GO enrichment analyses, 4511, 2645, and 4736 genes were respectively associated with biological process, cellular component, and molecular function terms (Supplementary Figure S2B). Further examination related to CDSs encoded by *P. silvicola* T1-3-2$^T$ was undertaken using KEGG pathway analyses. These analyses indicated that most gene annotations were associated with metabolism and environmental information processing pathways. Regarding metabolic pathway annotation, the terpenoids, polyketide metabolism pathways, and the biosynthesis of other secondary metabolites crucial for antibiotic synthesis were associated with 42 and 63 genes, respectively. Among these pathways, the terpenoid backbone biosynthesis (map00900), streptomycin biosynthesis (map00521), monobactam biosynthesis (map00261), acarbose and validamycin biosynthesis (map00525), and the biosynthesis of siderophore group nonribosomal peptides (map01053) were associated with 13, 10, 8, 7, and 7 genes, respectively (Supplementary Figure S2C).

## 2.7. Further Annotation of the Features of P. silvicola sp. nov.

Carbohydrate-active enzymes (CAZymes) play an important role in the metabolic processing of complex carbohydrates. Additionally, 235 putative CAZyme-encoding genes distributed unevenly among six families were identified within the genome of *P. silvicola* T1-3-2$^T$ (Table S5). These included 36 enzymes with auxiliary activities (AAs), 12 carbohydrate-binding modules (CBMs), 46 carbohydrate esterases (CEs), 4 polysaccharide lyases (PLs), 69 glycoside hydrolases (GHs), and 68 glycosyltransferases (GTs) (Supplementary Figure S2D; Table S5). The GH and CE classes of CAZymes included members with the potential to degrade cell wall components of plants and phytopathogens, including endo-1,4-β-glucanase (EC 3.2.1.4), 6-phospho-β-glucosidase (EC 3.2.1.86), α-glucosidase (EC 3.2.1.20), endoglucanases (EC 3.2.1.4), beta-1,3-glucan, oligoxyloglucan reducing end-specific cellobiohydrolase (EC 3.2.1.150) and xyloglucanase (EC 3.2.1.151) capable of degrading cellulose, and feruloyl esterase and acetyl xylan esterase (EC 3.1.1.72) of hemicellulose-degrading proteins (Table S6).

Moreover, five antimicrobial resistance (AMR) genes were found to be encoded in the *P. silvicola* T1-3-2$^T$ genome using CARD (https://card.mcmaster.ca/analyze/rgi/, accessed on 25 June 2021), including MexB (GE001972 and GE003456), MexF (GE004379), sdiA (GE005628), and MexS (GE006942). All five of these genes are members of the resistance-nodulation-cell division (RND) antibiotic efflux pump family, which can confer antibiotic resistance to both eukaryotic and prokaryotic cells by pumping antibiotics out of these cells. Notably, 2431 genes from *P. silvicola* T1-3-2$^T$ were annotated based on protein sequences from the VFDB (full virulence factor database; setB; http://www.mgc.ac.cn/VFs/search_VFs.htm, accessed on 24 December 2021), with 23 of these genes exhibiting high E-values associated with nine important *P. aeruginosa*-derived virulence factors. Moreover, 369 genes from the T1-3-2$^T$ strain were annotated in pathogenic *Pseudomonas* species in the host-pathogen interaction database (PHI0base), of which 238 and 114 were respectively similar to proteins encoded by *Pseudomonas aeruginosa* and *P. syringae*. At the same time, the remaining 18 were similar to proteins encoded by *P. cichorii* (8), *P. fluorescens* (2), and *P. savastanoi* (8).

## 2.8. Genomic Identification of Biocontrol Determinants in P. silvicola T1-3-2$^T$

Our previous research found that T1-3-2$^T$ with antibiotic functions against a range of phytopathogenic fungi [23] and genomic identification of biocontrol determinants was

analyzed. *Pseudomonas* species can synthesize broad-spectrum antibiotic compounds, including phenazine-1-carboxylic acid, 2,4-diacetylphloroglucinol, pyoluteorin, pyrrolnitrin, and protein-type compounds that can suppress pathogen growth [30,31]. In the T1-3-2[T] genome, 19 genes were annotated for antibiotics and secondary metabolites. Phenazine biosynthesis genes were identified, including five copies of phzS, four copies of phzH, two copies of phzD1, and one of the phzC1, phzE1, phzF1, phzG1, and phzR genes. Most phenazine-producing *Pseudomonas* spp. mediate phenazine biosynthesis by the conserved PhzB, PhzC, PhzD, PhzE, PhzF, and PhzG genes [32]. Some accessory genes, phzH, phzM, and phzS, encode phenazine-modifying enzymes in the strain-specific production of particular phenazine derivatives [33]. Congruously, phenazine and its derivatives were detected in the crude of fermentation by liquid mass spectrometry, including phenazine, 1, 6-dihydroxyphenazine, and 10-acetyl-3, 7-dihydroxyphenazine (unpublished data). Additionally, three genes encoding pyocin, pyocin R2, and the pyocin activator protein were detected in the T1-3-2[T] genome (Table S8). As reported, the production of siderophores confers antagonistic action on *Pseudomonas* species through the competition of available trace metals with phytopathogens and indirectly supports plant growth [34]. Ten genes associated with siderophore biosynthesis and uptakes were annotated together with 13 genes involved in pyoverdin biosynthesis in the T1-3-2[T] genome (Table S8). In addition, some genes were annotated with volatile organic compound production, of which 12 were required for HCN generation (hcnABC) (Table S8). Additionally, secondary metabolite gene clusters in the T1-3-2[T] genome included the Mangotoxin, Enterobactin, Vanchrobactin, and Kanamycin gene clusters with 71%, 16%, 20%, and 1% similarity, respectively (Table S7).

The PGP activity of the T1-3-2[T] strain was another important trait for its host, *C. laceolata* [23]. The activity of PGPP strains was often attributable to their abilities for nitrogen fixation, inorganic phosphate solubilization, ACC deaminase, and IAA production [35,36]. The biosynthesis proteins of pyrroloquinoline quinone (PQQ) were encoded by the PqqA, PqqB, PqqC, PqqD, and PqqE genes, which are related to phosphate solubilization [37]. Moreover, the ABC transport complex (PstABCS) with the PstA/B/C/S genes mediates the uptake of inorganic phosphate under conditions of phosphate starvation, as do the Pho regulon genes, including mediators of alkaline phosphatase activity [18]. The T1-3-2[T] strain harbored multiple gene clusters that mediate phosphate solubilization, including the PqqBCDEF genes, nine genes related to phosphate transport system permease (PstA, PstB, PstC, and PstS), and the Pho regulon gene clusters. Remarkably, most of these genes had two copies in the T1-3-2[T] strain, potentially conferring robust phosphate solubilization activity (Table S8). Bacteria capable of producing plant hormones can simultaneously promote root elongation and biomass production while circumventing host defense mechanisms via the depression of IAA signaling [38]. Reductions in ethylene levels through 1-aminocyclopropane-1-carboxylic acid (ACC) deamination can also promote plant growth [39]. The annotation of the T1-3-2[T] genome revealed genes associated with IAA production (three genes), nitrogen fixation (three genes), and ACC deaminase activity (two genes) (Table S8).

Meanwhile, several genes associated with plant-bacteria interactions were annotated within the genome of the T1-3-2[T] strain, including genes involved in chemotaxis, pilus assembly, and flagellar assembly (Table S8). The T1-3-2[T] strain also exhibited promising features, including the presence of multiple copies of genes coding for lytic enzymes, including two beta-glucosidase (bglX) genes, two 6-phospho-beta-glucosidase (BglB) genes, three 6-phospho-beta-glucosidase (AscB) genes, and four rhamnolipid production genes (Table S8). These enzymes play important roles in the inhibitory activity of *Pseudomonas* species in plants, exhibit a high degree of emulsification activity, and offer potential due to their ability to lower surface tension levels in the biocontrol of phytopathogens [10,40]. Strain T1-3-2[T] additionally encodes several catalases, superoxide dismutase [Fe/Mn], peroxidase, and thioredoxin/glutathione peroxidase genes involved in the protection of plants against damage. As such, these genomic sequencing and data mining efforts outlined

multiple potential biological and molecular mechanisms whereby *P. silvicola* T1-3-2$^T$ can promote plant growth and antagonize phytopathogens.

## 3. Conclusions

In summary, strain T1-3-2$^T$ has multiple polar flagella, a specific quinone system of methylnaphthoquinone MK8, and a 1.48 G + C mol% difference, which is distinct from its closest known relative "*P. eucalypticola*" through a polyphasic approach. We thus propose that T1-3-2$^T$ be designated as a novel species with the name *P. silvicola* sp. nov. Its ability to exert antifungal properties and PGP activity were most likely conferred by the genes associated with antibiotic and secondary metabolite production, such as the uptake and biosynthesis of phenazine biosynthesis, siderophores, pyoverdine biosynthesis and uptakes, and inorganic phosphate solubilization. These studies have laid the foundation for a basic understanding of T1-3-2$^T$. Yet, future investigations are required to elucidate its interaction with its host and identify the critical factors that govern its antibiotic production and secondary metabolism during fermentation. Such insights will be pivotal to facilitating and enhancing its development and application.

## 4. Description of *P. silvicola* sp. nov.

*Pseudomonas silvicola*, (sil.vi.co' la. Adj. silvicola, meaning living in or inhabiting forests or woods, in reference to the wooded area from which this species was isolated at the Lechang Longshan Forest Farm in Guangzhou, China).

*P. silvicola* colonies were beige and round with smooth edges on an LB medium at 25 °C. The cells of this species are gram-negative nonsporulating rods ~0.6–0.9 × 1.8–2.6 μm in size, with the motility of these microbes being facilitated by multiple polar flagella. The most prevalent fatty acid molecular esters (FAMEs) were $C_{16:0}$, $C_{17:0}$ cyclo, summed feature 8 ($C_{18:1}\omega7c/C_{18:1}\omega6c$: 14.71%), and summed feature 3 ($C_{16:1}\omega7c/C_{16:1}\omega6c$: 10.13%). The respiratory ubiquinone is Q-9. The G + C content of this strain is 61.65%.

The T1-3-2$^T$ strain exhibited nitrate reduction, arginine hydrolysis, and D-xylose acidification activity. The citrate, catalase, and arginine dihydrolase tests were all positive. This bacterium could not hydrolyze starch, Tween-80, or gelatin but could grow at temperatures from 4 to 37 °C, with optimal growth in the 25–30 °C range. The micro-organism can thrive in an LB medium with NaCl concentrations ranging from 0 to 2% (*w/v*) and pH values between 5.0 and 8.0. Optimal growth conditions include a 0.5% NaCl concentration and a pH of 7. Biolog GEN_III microplate analyses revealed strain T1-3-2$^T$ to be capable of oxidizing a range of carbon sources, including α-D-glucose, D-galactose, D-fucose, L-rhamnose, D-serine, L-alanine, L-aspartic acid, L-glutamic acid, D-galacturonic acid, L-galactonic acid lactone, D-gluconic acid, glucuronamide, mucic acid, quinic acid, D-saccharic acid, p-hydroxy-phenylacetic acid, L-lactic acid, citric acid, L-malic acid, γ-amino-butyric acid. Positive reactions were obtained for potassium tellurite, tetrazolium violet, tetrazolium blue, 1% sodium lactate, niaproof 4, rifamycin SV, lincomycin, and vancomycin, while weak positive reactions were obtained for D-trehalose, gentiobiose, D-melibiose, D-mannose, D-fructose,3-methyl glucose, L-fucose, D-mannitol, D-arabitol, glycerol, D-glucose-6-$PO_4$, D-fructose-6-$PO_4$, L-serine, L-arginine, D-malic acid, bromo-succinic acid, β-hydroxy-D,L butyric acid, α-keto-butyric acid, acetoacetic acid, acetic acid, formic acid, aztreonam, sodium bromate, nalidixic acid, fusidic acid, and guanidine HCl. T1-3-2$^T$ was found incapable of oxidizing other tested organic substrates.

The type strain, T1-3-2$^T$, was isolated from the cones of *Cunninghumia laceolata* at Longshan forest farm in Langtian Town, Lechang, Guangdong Province (131°16′58″, 45°40′53″). T1-3-2$^T$ was deposited in China Forestry Culture Collection Center (CFCC) (CFCC 16902) and Guangdong Provincial Microbial Culture Preservation Center (GDMCC) (GDMCC 1.3902). The accession numbers for the 16S rRNA gene and the draft genome sequence of the type strain are OM920535 and CP093280.

## 5. Methods

### 5.1. Bacterial and Fungal Growth Conditions

The 50 healthy and tender (1 month after flowering) cones of *Cunninghamia lanceolate* were randomly selected. All samples were washed with water and surface-sterilized with 75% ethanol for 1 min and 10% sodium hypochlorite for 5 min. The samples were then cut into pieces and washed with a sterile PBS solution. Finally, sterilized tissues were placed on the surface of a Petri plate with Luria–Bertani (LB) medium. The strain was purified and stored at −70 °C in a 30% glycerol suspension. For extracting the genome of T1-3-2$^T$, bacteria were cultured at 30 °C on Lysogeny broth medium (LB: 10 g peptone, 5 g yeast extract, and 5 g NaCl per 1000 mL deionized water, pH = 7.0) for 24 h. For use as a reference strain, the type strain of "*P. eucalypticola*" NP-1$^T$ was provided by Qinghua Zhang from the Institute of Forest Protection of the Forestry College of Fujian Agriculture and Forestry University.

### 5.2. PCR Amplification and DNA Sequencing

For DNA extraction, the T1-3-2$^T$ strain was inoculated and cultured in 200 mL of LB media for 24 h at 28 °C. Bacterial cells were harvested via centrifugation for 1 min at 12,000 rpm. The bacterial DNA was isolated using a Rapid Bacterial Genomic DNA Isolation Kit (B518225, Sangon Biotech, China). In order to establish the taxonomic placement of the T1-3-2$^T$, 16S rRNA partial sequence was amplified with the following primers: 27F (5′-AGAGTTTGATCCTGGCTCAG-3′) and 1492R (5′-ACGGCTACCTTGTTACGACTT-3′) [41]. Moreover, three gene sequences were amplified using the primers UP1E (5′-CAGGAAACAGCTATGACCAYGSNGGNGGNAARTTYRA-3′)/APrU(5′-TGTAAAACGA CGGCCAGTGCNGGRTCYTTYTCYTGRCA-3′) for the gyrB fragment [42], LAPS (5′-TGGCCGAGAACCAGTTCCGCGT-3′)/LAPS27 (5′-CGGCTTCGTCCAGCTTGTTCAG-3′) for the rpoB fragment [43], and 70F (5′-ACGACTGACCCGGTACGCATGTAYATGMGNGA RATGGGNACNGT-3′)/70R (5′-ATAGAAATAACCAGACGTAAGTTNGCYTCNACCATY TCYTTYTT-3′) for the rpoD fragment [42]. Each PCR reaction mixture contained 12.5 µL Taq PCR Mix (2X) (B110006, Diamond), 1 µL of each primer (10 µM), and 1 µL of template DNA for a total volume of 25 µL. The PCR amplification products were visualized by 2% agarose gel electrophoresis (60 V. 90 min). DNA sequencing was performed using an ABI PRISM® 3730XL DNA Analyzer with BigDye® Terminater Kit v.3.1 (Invitrogen) at Sangon Biotech (Shanghai) Co., Ltd. (Shanghai, China). All PCR production and the genome data were deposited in GenBank.

### 5.3. Phylogenetic Analysis

The 16S rRNA gene sequencing of T1-3-2$^T$ was performed via a similarity-based search against quality-controlled 16S rRNA gene sequences in the EzBioCloud database (https://www.ezbiocloud.net/, accessed on 16 January 2023) [44]. Then, the 16S rRNA gene sequences of T1-3-2$^T$, and the top-hit strains were used to construct a phylogenetic tree based on the neighbor-joining method with 1000 bootstrap replicates using MEGA 7 software [45]. For further analyses of T1-3-2$^T$, a multilocus sequence analysis (MLSA) with four concatenated genes (16S rRNA (1421 bp), gyrB (908 bp), rpoB (1123 bp), and rpoD (776 bp)) was performed. The remaining sequences included in this manuscript were obtained from public databases, and their accession numbers are listed in Supplementary Table S2. The phylogenetic MLSA tree was constructed using a maximum likelihood (ML) method [46]. ML was implemented on the CIPRES Science Gateway portal (https://www.phylo.org, accessed on 2 December 2022) using RAxML-HPC BlackBox 8.2.10 [47], employing a GTRGAMMA substitution model with 1000 bootstrap replicates. Phylogenetic trees were viewed using FigTree v.1.3.1 and processed with Adobe Illustrator CS5. The nucleotide sequence data for the new taxa were deposited in GenBank and are listed in Table S2.

### 5.4. Genome Sequencing and Analysis

The whole-genome DNA of T1-3-2[T] was extracted with an Ezup Column Bacteria Genomic DNA Purification Kit (B518225, Sangon Biotech, China), and sequenced using Oxford Nanopore technologies at Biomarker Technologies Co., Ltd. (Beijing, China). The low-quality reads were filtered using SMRT 2.3.0, and the filtered reads were then assembled with Canu v1.5 (https://github.com/marbl/canu, accessed on 2 June 2021). Gene prediction was performed using Prodigal software V2.6.3 (https://github.com/hyattpd/Prodigal/, accessed on 25 June 2021) [48]. Three types of rRNA in the genome were accurately predicted according to the covariance model using Infernal software V1.1.3 to [49]. Pseudogenes were predicted using the software of genBlastA v1.0.4 [50] and Genewise v2.2.0 [51]. Genomic islands, prophages, and gene clusters were predicted using the IslandPath-DIMOB v0.2 [29], Prophage PhiSpy v2.3 [52], and antiSMASH v5.0.0 softwares [53], respectively. The forecasted proteins were subjected to a blast (e-value: $1 \times 10^{-5}$) against various databases, including Nr [54], Swiss-Prot [55], TrEMBL25, KEGG [56], eggnog [57], and Blast2go [58] for functional annotation, with the last being utilized for GO [59] annotation. In addition, the pathogenic potential and drug resistance were investigated by executing a blast against the CAZy [60], CARD [61], PHI-base [62], and VFDB [63] databases.

The DNA G + C mol% was obtained based on genomic sequences. The similarity analyses, including ANI calculations based on BLAST+ (ANIb) or MUMmer (ANIm), and tetranucleotide signatures (TETRA) comparisons of the T1-3-2[T] genome to that of closely related *Pseudomonas* species ("*P. eucalypticola*": GCA_013374995, *P. coleopterorum*: GCA_900105555), and *P. rhizosphaerae*: GCA_000761155), were performed using the JSpecies software tool (http://jspecies.ribohost.com/jspeciesws, accessed on 28 April 2022). DNA-DNA hybridization (DDH) was calculated in silico using the GGDC 2.1 service with the BLAST+ method (http://ggdc.dsmz.de, accessed on 29 April 2022). DDH analyses were performed and analyzed with the "*P. eucalypticola*" (GCA_013374995), *P. coleopterorum* (GCA_900105555), and *P. rhizosphaerae* (GCA_000761155) reference genomes using the local alignment tool. DDH results were based on the recommended Formula 2, independent of genome length.

The phylogenomic tree inferred using FastME 2.1.6 from distances calculated via Genome BLAST Distance Phylogeny (GBDP) was constructed using Type (Strain) Genome Server (TYGS) web servers (https://tygs.dsmz.de/18, accessed on 29 April 2022). The resulting intergenomic distances were used to infer a balanced minimum evolution tree with branch support using FASTME 2.1.6.1 with SPR postprocessing. Branch support was inferred from 100 pseudo-bootstrap replicates each. The trees were rooted at the midpoint and visualized using PhyD3.

### 5.5. Morphological, Physiological, and Biochemical Tests

Gram staining was performed according to standard methods [64]. Cell morphology was examined via microscopy (Leica DM 2500), and flagellar arrangements were determined using transmission electron microscopy (TEM) after overnight incubation in a PDA medium at 25 °C. Growth at various temperatures (4, 10, 25, 30, 37, 41, 45, and 50 °C), in the presence of salt (NaCl, 0, 0.01, 0.025, 0.05, 0.1, 0.15, 0.20, 0.50, and 1–15% $w/v$), and at different pH values (3–12) was tested in LB media by monitoring the $OD_{600}$ after three days [27]. Gelatin hydrolysis was investigated using microbiochemical bacterial identification tubes (GB191, Hopebio). The experiments above were repeated in triplicate. According to the manufacturer's instructions, additional physiological and biochemical characteristics were determined using the Biolog GNIII microplate and API 20NE system.

### 5.6. Chemotaxonomic Analysis

Whole-cell fatty acid methyl esters (FAMEs) from *P. silvicola* T1-3-2[T] were tested under standardized conditions at the Guangdong Culture Collection, Guangzhou, China. The procedures for collecting, saponifying, methylating, and extracting cellular fatty acids

were carried out per the protocols outlined in the Sherlock Microbial Identification System (MIDI). Cellular fatty acid peaks, names, and percentages were analyzed using an Agilent 6890 N gas chromatograph, with the MIDI Microbial Identification System using the TSBA6 method and the Sherlock Microbial Identification software package version 6.138.

*5.7. Statements*

Experimental research and field studies were constructed in compliance with relevant institutional, national, and international guidelines and legislation. We have permission to collect any tissues of *Cunninghumia laceolata* at Longshan Forest Farm in Langtian Town, Lechang, Guangdong Province. All methods were carried out in accordance with the relevant guidelines, including the IUCN Policy Statement on Research Involving Species at Risk of Extinction and the Convention on the Trade in Endangered Species of Wild Fauna and Flora.

**Supplementary Materials:** The following supporting information can be downloaded at: https://www.mdpi.com/article/10.3390/f14061089/s1. Figure S1. Neighbor-joining phylogenetic tree based on concatenated 16S rRNA-gyrB-rpoB- rpoD sequences of T1-3-2$^T$ and the related *Pseudomonas* species. Figure S2. Clusters of Orthologous Groups of proteins (COGs) annotation (A). Clusters of Gene Ontology (GO) annotation (B). Clusters of KEGG annotation (C). Gene count distributions of carbohydrate-active enzyme families. GH glycoside hydrolases, GT glycosyl transferases, PL polysaccharide lyases, CE carbohydrate esterases, CBM carbohydrate-binding modules, AA auxiliary activities (D). Table S1. Accession numbers were used in phylogenetic analysis based on the 16S rRNA. Table S2. Accession numbers of the sequences of the type strains of different *Pseudomonas* species used in the MLSA phylogenetic analysis. Table S3. Type strains and their genome sources in the phylogenomic tree. Table S4. Functional categories bases on eggNOG of the protein-coding genes of the *Pseudomonas* silvicola sp. nov. Table S5. Genes related to carbohydrate-active enzymes (CAZymes) in *Pseudomonas silvicola* sp. nov. Table S6. CAZymes involved in plant and fungal cell wall degradation identified in *Pseudomonas silvicola* sp. nov. genome. Table S7. Secondary metabolite gene clusters identified in the *Pseudomonas silvicola* sp. nov. using antiSMASH v. 5.0.0. Table S8. Genes attributed to biocontrol, plant growth promotion and colonization traits identified in *Pseudomonas* sp. T1-3-2$^T$ strain genome.

**Author Contributions:** L.T. and Y.Z. conceived and conducted the experiments. L.T., Y.Z., H.Y., Q.Z., C.Q. and H.Q. analyzed and confirmed the results. L.T. and Y.Z. took the lead in writing the manuscript under the supervision of J.X. and C.Q. All authors have read and agreed to the published version of the manuscript.

**Funding:** This study is funded by Guangdong Basic and applied basic research Foundation (Youth Project) (project No.: 2019A1515110596) and the Guangdong Forestry Science and Technology Innovation Project (project No.: 2023KJCX020). And the APC was funded by the Guangdong Forestry Science and Technology Innovation Project (project No.: 2023KJCX020).

**Institutional Review Board Statement:** Not applicable.

**Informed Consent Statement:** Not applicable.

**Data Availability Statement:** The datasets generated and analyzed during the current study are available in the NCBI repository (https://www.ncbi.nlm.nih.gov/nuccore/CP093280.1/, accessed on 12 March 2022).

**Acknowledgments:** The authors thank Ning Jiang from the Chinese Academy of Forestry, Beijing, China, for the helpful suggestions and revision of this manuscript. The authors thank Qinghua Zhang from the institute of forest protection in college of Fujian agriculture and forestry university, Fuzhou, China, for providing the type strain of "*P. eucalypticola*" NP-1$^T$. The authors would like to thank all the reviewers who participated in the review and MJEditor (www.mjeditor.com, accessed on 17 May 2023) for its linguistic assistance during the preparation of this manuscript.

**Conflicts of Interest:** The authors declare no conflict of interest.

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
