# Peer review of "Taxonomic Description and Complete Genome Sequencing of Pseudomonas silvicola sp. nov. Isolated from Cunninghamia laceolata"

_forests, doi:10.3390/f14061089_

Round 1

Author Response

Dear Editor and reviewer,

Thanks for your revisions and suggestions. And my responses are as follows:

Introduction

Comment 1: The study has no mechanistic hypothesis. The significance of the study and a solid hypothesis of the present study at the end of the introduction should be provided.

R: Added. Please see text in introduction part. ‘In our previous study….which forming a monophyletic group with Pseudomonas eucalypticola based on 16s rRNA phylogenetic analyses. However, its taxonomic classification and potential utility were not clarified clearly, and this hindered its fundamental research and utilization.’

Comment 2: Should be add one paragraph about effect of Pseudomonas species on antibiotic functions against a range of phytopathogenic fungi and its PGP traits.

R: Added the text “Many beneficial plant-associated ……. engagement of more robust systemic resistance9, 10

Comment 3: Should be add one paragraph about Cunninghamia lanceolate.

R: Added the text including “Cunninghamia lanceolate (Lamb.) Hook, …. substantial tree damage and loss in China.”

Comment 4: Authors should be provided full method about endophytic bacterial strain isolated from Cunninghamia laceolata.

R: Added the text, such as“The 50 healthy and tender (1 month after flowering) cone …..of a Petri plate with Luria–Bertani (LB) medium.”

Discussion

Comment 5: In the discussion section, is needed an explanation about how Pseudomonas species of antibiotic functions against a range of phytopathogenic fungi.

R: We had adjusted this part, and the first paragraph of discussion is the answer of this question. Please see it in the revised manuscript.

Comment 6: What are the mechanisms that allow Pseudomonas species to antifungal properties against phytopathogens in the Colletotrichum, Pestalotiopsis, Fusarium, and Phellinus genera.

R: According to the results, the mainly mechanisms allowed T1-3-2T against phytopathogens are concluding these three parts: 1. There are phenazine and phenazine derivatives were detected in the crude extraction. Meanwhile, Phenazine biosynthesis genes were identified including 5 copies of phzS, 4 copies of phzH, 2 copies of phzD1, 1 of phzC1, phzE1, phzF1, phzG1, and phzR genes in the genome of T1-3-2T. 2. Besides, secondary metabolite gene clusters, such as Mangotoxin, Enterobactin, Vanchrobactin, and Kanamycin gene clusters were showed in T1-3-2T genome. 3. The production of siderophores was proved in the T1-3-2T by CAS medium, and genes associated with siderophore biosynthesis and uptakes were annotated in T1-3-2T genome.

Comment 7: The conclusion needs to be rewritten.

R: It has been rewritten.

Reviewer 2 Report

Reviewer

Comments for the author

The manuscript title is "Taxonomic description and complete genome of Pseudomonas silvicola sp. nov., a candidate biocontrol agent isolated from Cunninghamia laceolata". By Longyan Tian. The present study was designed to conduct polyphasic taxonomic analysis and genome-based studies. After carefully reviewing this manuscript, I found tonnes of fundamental, grammatical, and typographical errors. So, the current form is not up to the scientific standard. So, I recommend a resubmission.

Major points

  1. Please revise the manuscript title; the current form is very hard to understand.
  2. The entire abstract is poorly written and not up to the scientific standard. Please add the following information clearly; background information on the study, objectives, methodology, results, and a conclusion. The current form is very hard to understand.
  3. The introduction part is not sufficient; please write elaborately with recent information.
  4. In the results section, the morphological information about the bacteria is missing; please add it.
  5. The quality of the figures 3A, B, C, and D  is poor, so please replace them with high-quality images.
  6. In Table 2, what are the red colour marks?
  7. The entire results and discussion part looks like a textbook; the current format is very hard to follow, so please revise it.
  8. The conclusion part is not sufficient, so please write elaborately.
  9. Line 345: What is the source of bacterial culture?
  10. The entire manuscript is not up to the scientific standard, so before resubmitting it, please carefully revise it.

The manuscript English language is poor and not up to the scientific standard so please edit language with English expert or native persons.

Author Response

Dear reviewer,

Thanks for your revisions and suggestions. And my responses are as follows:

Reviewer 2

Major points

  1. Please revise the manuscript title; the current form is very hard to understand.

R: Revised.

  1. The entire abstract is poorly written and not up to the scientific standard. Please add the following information clearly; background information on the study, objectives, methodology, results, and a conclusion. The current form is very hard to understand.

R: Revised and added.

  1. The introduction part is not sufficient; please write elaborately with recent information.

R: Added the related information.

  1. In the results section, the morphological information about the bacteria is missing; please add it.

R: The morphological information was described in the 2.3, including “The cells of this species are Gram-negative non-sporulating rods ~0.6-0.9 × 1.8-2.6 μm in size, with the motility of these microbes being facilitated by multiple polar flagella ~2.0-5.2 μm in length (Fig. 3D and E).

  1. The quality of the figures 3A, B, C, and D is poor, so please replace them with high-quality images.

R: Replaced.

  1. In Table 2, what are the red colour marks?

R:The blank has been added.

  1. The entire results and discussion part looks like a textbook; the current format is very hard to follow, so please revise it.

R: Revised.

  1. The conclusion part is not sufficient, so please write elaborately.

R: Revised.

  1. Line 345: What is the source of bacterial culture?

R: Added the source of bacteria. Please see in text ‘The 50 healthy and tender (1 month after flowering) cone of Cunninghamia lanceolate……. Strain was purified and stored at -70℃ in a 30% glycerol suspension. ’

  1. The entire manuscript is not up to the scientific standard, so before resubmitting it, please carefully revise it.

 R: Revised.

The manuscript English languages is poor and not up to the scientific standard so please edit language with English expert or native persons.

 R: The English language revised by the MJEditor.

Reviewer 3 Report

General comments

The introduction must be improved. You need to provide all the background information to the readers to understand the scope of your research. Moreover, it would be nice to clearly state the aims and the objectives of this work.

Unless it is not obvious PGP and antifungal properties that you claim are not obvious in this paper. You claim that they were proved in previous work, but you need to demonstrate it. So, explain the previous work and refer to it. Moreover, in order to prove the PGP you need extensive experimental work by measuring the vegetative characteristics of the plants which is missing from this work. The same applies to the antifungal properties. You need to prove it as a response of the plants under glasshouse and/or field conditions.

It is good and interesting work but I feel that you tend to overestimate the important findings. You have proven that you found a new strain and this strain has the potential to be used as a PGP but this needs to be proved by experimentation.

It would be nice to provide some future plans/work for the applications of your findings.

Specific comments

Lines 44-45: "The Pseudomonas genus…. substantial metabolic versatility”. Ok, give us some examples.

Lines 46-48: “In our previous study…. Fusarium, and Phellinus genera”. Based on what did you chose cones of Cunninghamia lanceolate and why did you decide to test it as PGP?

Lines 304-306: I do not think that you have provided sufficient evident for such claim. Your research involved only the identification, the characterization of the strain and in-vitro tests. In order to claim that the strain can be used a biocontrol agent you need extensive and detailed experimentation in glasshouse (seedlings) and in field conditions.

The manuscript is well written in the English language. Minor editing of English language required.

Author Response

Dear reviewer,

Thanks for your revisions and suggestions. And my responses are as follows:

Reviewer 3

Comments and Suggestions for Authors

General comments

The introduction must be improved. You need to provide all the background information to the readers to understand the scope of your research. Moreover, it would be nice to clearly state the aims and the objectives of this work.

R: Revised and added the information about background, aims and the objectives in the revised manuscript.

Unless it is not obvious PGP and antifungal properties that you claim are not obvious in this paper. You claim that they were proved in previous work, but you need to demonstrate it. So, explain the previous work and refer to it. Moreover, in order to prove the PGP you need extensive experimental work by measuring the vegetative characteristics of the plants which is missing from this work. The same applies to the antifungal properties. You need to prove it as a response of the plants under glasshouse and/or field conditions.

R: The PGP activity and antifungal properties had been proved by measuring the vegetative characteristics of its host and efficiency for controlling in anthracnose of Chinese fir in our previous work. Revised and added our previous work in the revised manuscript. See it in revised munucsript-‘In our previous study, the endophytic strain T1-3-2 isolated from the cones of Cunninghamia lanceolate have been found to exhibit plant growth promotion (PGP) and antifungal properties against phytopathogens in the Colletotrichum, Pestalotiopsis, Fusarium, and Phellinus genera23

It is good and interesting work but I feel that you tend to overestimate the important findings. You have proven that you found a new strain and this strain has the potential to be used as a PGP but this needs to be proved by experimentation.

R: As I mentioned above, the PGP activity and antifungal properties had been proved in previous work. This manuscript was intended to explain what the taxonomic position and biocontrol molecule potentials of this strain, but due to the poor expression made confusion to reviewer.

It would be nice to provide some future plans/work for the applications of your findings.

R: Added future plans in the conclusion.

Specific comments

Lines 44-45: "The Pseudomonas genus…. substantial metabolic versatility”. Ok, give us some examples.

R: Added examples in introduction. ‘Many beneficial plant-associated Pseudomonas species …. P. oryziphila 1257 6, to offer substantial genetic promise as potential biocontrol agents.’included.

Lines 46-48: “In our previous study…. Fusarium, and Phellinus genera”. Based on what did you chose cones of Cunninghamia lanceolate and why did you decide to test it as PGP?

R: The cone of Cunninghamia lanceolate were randomly selected 50 healthy and tender (1 month after flowering).

Initial, we just test its biocontrol efficiency. Unexpectedly, the treatment just sprayed with the suspension of T1-3-2 showed PGP activity when compared with the control sprayed with sterile water. So, we conducted other tests to prove this trait in our previous work. 

Lines 304-306: I do not think that you have provided sufficient evident for such claim. Your research involved only the identification, the characterization of the strain and in-vitro tests. In order to claim that the strain can be used a biocontrol agent you need extensive and detailed experimentation in glasshouse (seedlings) and in field conditions.

R: We have rewritten this part.

Round 2

Reviewer 2 Report

The current form of the manuscript worthy for publication. All the very best.

English language is fine but also please carefully chech the spelling.